

# Validity and reliability of a teledentistry survey among dental practitioners in Saudi Arabia

Alla Alsharif[1], Doaa Felemban[2], Hala Bakeer[3] and Saba Kassim[4]

[1] Department of Preventive Dental Sciences, Taibah University College of Dentistry, Al-Madinah Al-Munawwarah, Saudi Arabia
[2] Department of Oral Basic and Clinical Sciences, Taibah University College of Dentistry, Al-Madinah Al-Munawwarah, Saudi Arabia
[3] Neelain University, Khartoum, Sudan
[4] Department of Preventive Dental Sciences, Taibah University, Al-Madinah Al-Munawwarah, Saudi Arabia

## ABSTRACT

**Background:** The perception of teledentistry use among dental practitioners in various contexts was assessed using the Teledentistry Survey (the TDS-24). However, this survey's psychometric analyses have not yet been analysed. This present study aims to examine the validity and reliability of the TDS-24 in a sample of dental practitioners in Saudi Arabia.

**Methods:** A self-administered questionnaire, including sociodemographic characteristics and the TDS, was distributed as a cross-sectional survey to 800 current dental practitioners in Saudi Arabia recruited *via* convenience and snowball sampling. The construct validity and reliability of the TDS were assessed using exploratory factor analysis (EFA) and Cronbach's alpha.

**Results:** The EFA of the survey yielded 20 items supporting a four-factor structure as follows: factor I (10 items), factor II (four items), factor III (three items) and factor IV (three items). The overall Cronbach's alpha was 0.85, while it was 0.86 for the first factor, 0.70 for the second factor, 0.52 for the third factor and 0.57 for the fourth factor.

**Conclusions:** The TDS-20, after excluding four items, revealed four factors with adequate psychometric properties, making it a valid and reliable tool in assessing teledentistry perceptions among dental practitioners in this study sample.

# INTRODUCTION

Teledentistry uses information technology and telecommunications for oral healthcare delivery, consultation, and education (*Di Cerbo & Morales, 2015*; *Scott et al., 2018*; *Yang et al., 2015*; *Daniel & Kumar, 2014*). It includes diagnosing and assessing the severity of dental conditions, treatment planning and scheduling, providing oral health education and reviewing cases (*Kumar et al., 2019*; *Jampani et al., 2011*). This technological approach allows underserved patients in rural areas to connect and communicate with dental service

Corresponding author
Alla Alsharif,
dr-alsharif@hotmail.com

providers, granting them access to general dentists and specialised dental treatments if needed (*Daniel, Wu & Kumar, 2013*; *Estai, Kruger & Tennant, 2016a*, *2016b*). Teledentistry has shown potential in fields such as oral medicine, orthodontics, dental trauma, periodontology and caries diagnosis in pediatric dentistry (*Haron et al., 2017*; *Batham, Pereira Kalia & Dilliwal, 2014*; *Marino et al., 2017*; *Bhambal, Saxena & Balsaraf, 2010*; *Morosini et al., 2014*). This cybernated technology has gained significant traction since COVID-19 pandemic onset. Teledentistry is one of the practical methods used extensively to recompense the difficulties of dental care accessibility. However, its application and practitioner interest in this regard have been subject to scrutiny (*Aboalshamat, 2020*; *Giraudeau, 2021*; *Macapagal, 2020*; *Abbas et al., 2020*; *Sycinska-Dziarnowska et al., 2021*; *Imran Farooq, Moheet & AlHumaid, 2020*; *Chaudhary et al., 2021*). Some studies state the positive reliability, validity, and efficacy on the usage outcomes (*Tiwari et al., 2022*; *Estai, Kruger & Tennant, 2016c*; *Aboalshamat et al., 2022*; *Alsharif & Al-harbi, 2020*). However, there are some factors such as information confidentiality related to patient data, informed consent, the risk of misdiagnosis, ethical factors, and financing problems that play a role in its impediment (*Aboalshamat et al., 2022*). To assess dental practitioners' perceptions of teledentistry, *Mandall, Quereshi & Harvey (2005)* developed the Teledentistry Survey (TDS) based on interviews with eight general dental practitioners (GDPs). The survey questions, which were derived from these interviews, focus on four factors: efficiency, usefulness, GDPs' perspectives and concerns. Although this survey has been widely used to study dental practitioners' perceptions of teledentistry in Australia, Canada, the United Kingdom, Saudi Arabia and Indonesia, limited evidence exists regarding its validity (*Estai, Kruger & Tennant, 2016a*; *Aboalshamat, 2020*; *Tiwari et al., 2022*; *Estai, Kruger & Tennant, 2016c*; *Aboalshamat et al., 2022*; *Alsharif & Al-harbi, 2020*; *Mandall, Quereshi & Harvey, 2005*; *Soegyanto et al., 2022*; *Patel & Antonarakis, 2013*). Although, teledentistry has been extensively covered in the region, this study will provide a unique perspective by focusing on the psychometric analysis of the TDS that hasn't been explored comprehensively in diverse contexts. Notably, the Saudi dental community comprises a diverse spectrum of dental professionals with expertise spanning many aspects of oral healthcare. The diversity of the dental field in Saudi Arabia enables our study to produce results that can be compared effectively with those of other countries using similar survey instruments and methodologies. Comparative research adds value to our work since it allows us to draw meaningful insights, benchmark our findings against international standards, and gain a deeper understanding of dental practices and trends globally. To establish the generalisability of the (TDS-24) in terms of assessing dental practitioners' perception of teledentistry, this study examined the factorial structure and psychometric properties (validity and reliability) of the TDS-24 among dental practitioners in Saudi Arabia. In our exploratory factor analysis of the TDS-24 survey, we aimed to measure its psychometric properties. This involved evaluating the factor loadings of each item to determine their strength and direction of association with the underlying factors. By identifying these factors and their corresponding items, we intended to provide insight into the latent constructs or domains within the survey and explore the potential reduction of the scale to a more concise and psychometrically sound version.

## MATERIALS AND METHODS

### Study design and sampling

This cross-sectional study used convenience snowball sampling technique. The researchers leveraged their professional networks and affiliations with dentists to recruit participants from various regions in Saudi Arabia. The research survey was distributed using social network sites, such as Twitter, Facebook, LinkedIn, WhatsApp, *etc*. The study included dentists who were actively practicing in the Kingdom of Saudi Arabia, who provide direct patient care and had various levels of experience. Dentists who were not practicing at the time of the study were excluded.

### Data collection and measures

An anonymous pretested English self-administered questionnaire, which took approximately around 15–20 min to complete and was made available for 1 month, was sent to participants using a direct link to a Google form *via* software application HTML (Hypertext Mark-up Language). The abovementioned social media platforms were used to distribute the Google form questionnaires. Upon the completion of the survey, the respondents were instructed to submit the web form, which then stored the data in an Excel spreadsheet specifically created for data storage and retrieval for analysis purposes.

The structured questionnaire survey included questions related to sociodemographic information (*e.g.*, age and gender), qualifications (*i.e.*, years of work experience in dentistry) and the TDS-24. The teledentistry survey used in this study was originally developed by *Mandall, Quereshi & Harvey (2005)*, *Estai, Kruger & Tennant (2016c)*. The Estai version has been employed in several studies and has been shown to be a more effective way of capturing all dentists' perspectives regarding teledentistry instead of focusing solely on orthodontics. Hence, this study employed the modified version of the survey, as developed by *Estai, Kruger & Tennant (2016c)*. This comprises four factors with a total of 24 items (Table 1). The first factor contained seven questions focusing on the perceived usefulness of teledentistry, the second factor included five questions assessing the efficiency of teledentistry, the third factor comprised seven questions examining the benefits of teledentistry for patients and fourth factor involved five questions related to the practice-related use of information technologies. Participants responded to all the questions in these factors using a five-point Likert scale ranging from one (strongly disagree) to five (strongly agree).

There is no consensus regarding the sample size needed to validate a survey that assesses various aspects of professionals' perspectives on using information and communication technology for health service provision. Thus, the sample size required to validate the TDS was calculated as 460. This size was based on the available recommendations (*i.e.*, the subject-item ratio (2–20 subjects per item) needed to perform the factor analysis) (*Kline, 1979*; *Hair et al., 1995*). However, 800 TDS questionnaires were distributed after accounting for non-responses and missing data (*Anthoine et al., 2014*).

### Statistical analyses

The Excel spreadsheet was exported into statistical software programs for analysis. Psychometric properties tests were performed to evaluate the construct validity and

**Table 1 Teledentistry survey TDS-24 items.**

| No. | Items |
| --- | --- |
| **DENTISTS' PERCEPTIONS OF USEFULNESS OF TELEDENTISTRY SYSTEM** | |
| 1 | Teledentistry would provide adequate diagnostic information |
| 2 | Teledentistry would be too expensive to set up |
| 3 | Teledentistry would save time as compared with a referral letter |
| 4 | Teledentistry would necessitate an additional appointment for taking photographs |
| 5 | Teledentistry would increase surgery time spent with the patient |
| 6 | Teledentistry would reduce costs for dental practices |
| 7 | Teledentistry would enhance clinical training and continuing education |
| **DENTISTS' PERCEPTION OF EFFICIENCY OF TELEDENTISTRY SYSTEM** | |
| 8 | Teledentistry would make the referral of new patients more efficient |
| 9 | Teledentistry would improve communications between dentists |
| 10 | Teledentistry would enhance guidance and advice |
| 11 | Teledentistry would help shorten waiting lists |
| 12 | Teledentistry diagnosis based on intra-oral images is as accurate as in a traditional clinical setting |
| **DENTISTS' PERCEPTION OF BENEFITS OF TELEDENTISTRY SYSTEM FOR PATIENTS** | |
| 13 | Teledentistry would be useful for patients in distant or rural locations |
| 14 | Teledentistry would be convenient for patients and well received by patients |
| 15 | Teledentistry would be helpful in monitoring a patient's condition |
| 16 | Teledentistry would help reduce unnecessary travel to hospitals |
| 17 | Teledentistry would help with patient information and education |
| 18 | Teledentistry would improve interaction and communication with patients |
| 19 | Teledentistry would save money for patients |
| **DENTISTS' CONCERN ABOUT PRACTICE-RELATED USE OF INFORMATION TECHNOLOGIES** | |
| 20 | Reliability of equipment |
| 21 | Technical incompatibility |
| 22 | Patient confidentiality when images are sent online to the hospital |
| 23 | Potential for tampering with computer images |
| 24 | Gaining patient consent for referrals *via* email |

reliability of the TDS. The construct validity of the TDS was assessed using exploratory factor analysis (EFA) rather than confirmatory factor analysis, as this was the first study to explore the TDS's structure. The conventional tests to determine the appropriateness of the dataset to undergo EFA were performed, and these included the following: a Kaiser–Meyer–Olkin Test (KMO) value of 0.6 or above (*Kaiser, 1974*; *MacCallum et al., 1999*), a significant Bartlett's Test of Sphericity value ($p \leq 0.05$) and a correlation matrix with many coefficients of three and above (*Tabachnick & Fidell, 2001*). This was followed by an assessment of the factorability of the components using principal component analysis with varimax rotation. Loading criteria for factor(s) of ≥0.50 were accepted to ease the interpretation of the EFA. *Comrey & Lee (1992)* suggested using the following cut-offs to assess item loadings: 0.32 poor, 0.45 fair, 0.55 good, 0.63 very good and 0.71 excellent. The number of factors was determined based on the yield of Cattell's scree test plotting

each of the factors against an associated eigenvalue that exceeded one. Regarding the reliability of the TDS, the internal consistency of the survey and subfactor extracted were tested using the Cronbach's alpha coefficients. In addition, the item-total correlation was calculated, and the impact of removing an item on the reliability of the survey was assessed. The statistical software program SPSS Version 21.0 was used for the EFA. A $p$-value < 0.05 was considered significant.

## Ethical approval

Ethical clearance to conduct this study was obtained from the Research Ethical Committee at Taibah University College of Dentistry (TUCDREC/20190918/ATAlSahrif). The study was conducted in accordance with the principles of the World Medical Association Declaration of Helsinki. A cover page preceded the questionnaire summarised the study aim and voluntary and anonymous nature of participation and provided the researchers' contact details for any research-related queries. In addition, an electronic informed consent statement was to be completed by all participants who agreed to participate. Otherwise, their questionnaires will not be opened. Electronically stored data was restricted to authorized personnel only and was stored in a secure manner. In order to maintain participant privacy and confidentiality, all personal identifiers were removed from the survey, and the data were anonymised during analysis. To enhance the dataset's reliability, preprocessing techniques, including data cleaning and validation, were applied. In order to protect participants' rights and ensure responsible handling of sensitive information, strict adherence to ethical and legal standards was maintained throughout the study.

# RESULTS

## Sample characteristics

A total of 800 survey questionnaires were sent out, of which 620 useful and completed surveys were analysed. The demographic characteristics of the respondents were analysed. A total of 373 individuals (62.2%) fell within the age range of 20–34 years, while 415 individuals (66.9%) were Saudi nationals. Additionally, 383 individuals (62%) were female. In terms of profession, 354 participants (57.1%) were classified as general dentists, 148 (23.9%) as dental specialists and 118 (19%) as consultants (Table 2). Among the respondents, 363 individuals (58%) were employed in the public sector, while 150 (24%) were engaged in dual practice, working in both the private and public sectors. Furthermore, 459 respondents (74%) had clinical experience of up to 10 years, while 161 (26%) had 11 or more years of experience (Table 2).

## Psychometric properties of the TDS

The factor analysis conducted for the entire dataset revealed that the KMO measure of sampling adequacy was 0.900. The dataset revealed a correlation matrix of many coefficients of ≥0.3 and Bartlett's test of sphericity was significant ($\chi^2$(df) = 3,863.94 (253); $p \leq 0.001$), supporting the suitability of the data for factor analysis. The square root of the average variance extracted (AVE) was compared with correlations between the constructs

**Table 2 Socio-demographic characteristics of the entire sample ($n$ = 620).**

| Variable | | Whole N (%) |
|---|---|---|
| Age (years) | 20–34 years | 373 (62.2) |
| | 35–44 years | 141 (22.7) |
| | ≥45 | 106 (17.1) |
| Nationality | Non-Saudi | 205 (33.1) |
| | Saudi | 415 (66.9) |
| Qualification | General dentist | 354 (57.1) |
| | Dental specialist | 148 (23.9) |
| | Dental consultant | 118 (19.0) |
| Work experience in dentistry | ≤10 years | 459 (74.0) |
| | ≥11 years | 161 (26.0) |
| Work setting | Private | 108 (17.4) |
| | Public | 362 (58.4) |
| | Both | 150 (24.2) |

under study to ensure discriminant validity. For each construct, the square root of AVE was larger than correlations with other factors. A four-factor solution accounting for 47.34% of variance was obtained from the factor analysis of the TDS. The slope of the Cattell's Scree plot test explicitly demonstrated the existence of these four factors (Fig. 1).

As shown in Table 3, most items had loading values higher than 0.50 on all factors. However, four items of the proposed TDS were dropped in this dataset analyses, namely items 12, 16, 21 and 24 (Table 3). Interestingly, some items demonstrated significant loading values on factors different from those originally assigned. For instance, items 8, 9 and 10 of factor II had loading on factor I (0.763, 0.702 and 0.671, respectively), and items 13, 15, 17 and 18 of factor III exhibit loading on factor I. Similarly, item 6 of factor I had loading on factor II (0.673); items 14, 16, and 19 of factor III had loading on factor II; items 2, 4 and 5 of factor I had loading on factor III (0.637, 0.621 and 0.681, respectively) and items 20, 22 and 23 of factor IV showed loading on factor IV. Some items were loaded on different factors from those originally considered. The Cronbach's alpha for all factors was 0.85, and it was 0.86 for the efficiency factor, 0.70 for the cost factor, 0.52 for the capability factor and 0.57 for the security factor (Table 3). The Cronbach's Alpha if items were deleted was >0.80 for each individual item, ranged from 0.841 to 0.859. For the corrected item-total correlation, most items had values higher than 0.50. More details are presented in Table 3.

## DISCUSSION

This is the first study to assess the factorial structure and psychometric properties of the TDS-24 (*Estai, Kruger & Tennant, 2016c*) using EFA (a data-driven method) based on a diverse sample of dental practitioners in Saudi Arabia. In comparison with Estai's proposed factors, our EFA found a well-separated four-factor structural model, the items

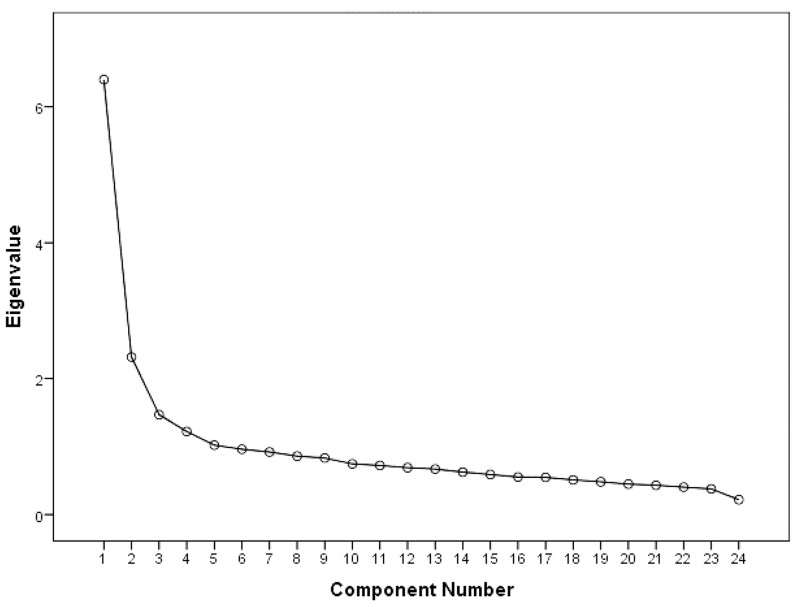

**Figure 1 TDS scree plot.**

of the four-factor model have greater loadings for their corresponding factor and almost half of the items can be explained by a single factor. It is a shortcoming of factor IV that it has only three items; however, according to the guidelines, one factor should have more than two items if possible (*Crawford, 1975*; *Zwick & Velicer, 1986*). The phenomenon in which some items show loadings on factors different from their intended ones is a common occurrence in factor analysis and can be attributed to ambiguity in item wording, item redundancy, conceptual overlapping or, in some cases, factors that are not strongly distinct. Also, the underlying structure of the data may not be perfectly aligned with the proposed factor model. On the other hand, the characteristics of the sample used in the factor analysis can influence the results. If the sample has unique traits or experiences that differ from the population the scale was designed for, this may lead to unexpected item-loadings in the factor analysis. Thus, researchers should explore the theoretical underpinnings of the scale, review the wording of ambiguous items and evaluate the conceptual overlap between factors. It might also be beneficial to conduct further analyses, such as confirmatory factor analysis, to test the stability and validity of the factor structure with an independent dataset. Ultimately, the goal is to ensure that the factor structure accurately represents the underlying constructs being measured and that items are appropriately assigned to the intended factors.

Four items (items 12, 16, 21 and 24) were excluded, and the exclusion of these items could have been due to their weak correlation with the underlying factors or their redundancy with other items. Thus, the exclusion of these items should be carefully considered when interpreting the results.

The EFA of the scale yielded 20 items supporting a four-factor structure as follows: factor I (10 items), factor II (four items), factor III (three items) and factor IV (three items). The differences in item loadings were anticipated. It is not realistic to expect that the factor

**Table 3 Reliability and validity psychometric analyses of the teledentistry survey.**

| Teledentistry survey factors | Corrected item-total correlation | Cronbach's alpha if item deleted | Factors loadings | | | |
|---|---|---|---|---|---|---|
| | | | Factor 1 | Factor 2 | Factor 3 | Factor 4 |
| **Efficiency in patient care** | | | | | | |
| 9. Improve communications between dentists | 0.565 | 0.842 | **0.763** | | | |
| 8. Make the referral of new patients more efficient | 0.579 | 0.841 | **0.702** | | | |
| 10. Enhance guidance and advice | 0.537 | 0.843 | **0.671** | | | |
| 7. Enhance clinical training and continuing education | 0.526 | 0.843 | **0.665** | | | |
| 3. Save time as compared with a referral letter | 0.560 | 0.842 | **0.673** | | | |
| 17. Help with patient information and education | 0.522 | 0.844 | **0.586** | | | |
| 7. Provide adequate diagnostic information | 0.530 | 0.842 | **0.541** | | | |
| 15. Helpful in monitoring a patient's condition | 0.506 | 0.844 | **0.568** | | | |
| 13. Useful for patients in distant or rural locations | 0.572 | 0.841 | **0.563** | | | |
| 18. Improve interaction and communication with patients | 0.527 | 0.843 | **0.538** | | | |
| **Cost reduction** | | | | | | |
| 6. Reduce costs for dental practices | 0.407 | 0.847 | | **0.673** | | |
| 19. Save money for patients | 0.394 | 0.847 | | **0.599** | | |
| 16. Help reduce unnecessary travel to hospitals | 0.545 | 0.842 | | <0.50 | | |
| 14. Convenient for patients and well received by patients | 0.542 | 0.842 | | **0.546** | | |
| 11. Help shorten waiting lists | 0.488 | 0.844 | | **0.559** | | |
| **Capabilities to improve practice** | | | | | | |
| 5. Increase surgery time spent with the patient | 0.227 | 0.855 | | | **0.681** | |
| 4. Necessitate an additional appointment for taking photographs | 0.266 | 0.853 | | | **0.621** | |
| 2. Too expensive to set up | 0.095 | 0.859 | | | **0.637** | |
| 12. Diagnosis based on intra-oral images is as accurate as in a traditional clinical setting | 0.356 | 0.849 | | | <0.50 | |
| **Security and confidentiality** | | | | | | |
| 20. Reliability of equipment | 0.179 | 0.855 | | | | **0.671** |
| 21. Technical incompatibility | 0.517 | 0.843 | | | | <0.50 |
| 22. Patient confidentiality when images are sent online to the hospital | 0.301 | 0.851 | | | | **0.684** |
| 23. Potential for tampering with computer images | 0.137 | 0.858 | | | | **0.739** |
| 24. Gaining patient consent for referrals *via* email | 0.517 | 0.843 | | | | <0.50 |

**Note:**
The bold values indicate the loading rating, with 0.55 being categorized as good, 0.63 as very good, and 0.71 as excellent.

loadings will be identical across countries. Dental practitioners from Saudi Arabia have perceived and reported items differently from those in other countries, which resulted in slight variability in the TDS loadings and, as such, factor naming. All the extracted factors showed acceptable or good internal consistency (Cronbach's α). Using such a model ensures sample adequacy and the validity and reliability of the interpreted data.

The EFA confirmed that the TDS-20 had good model fit. As this study was the first to assess the factorial structure and psychometric properties of the TDS-23, comparisons of

the findings of this study with those of other studies were not possible. The 20 items loaded under the four-factorial structure were suggested to be related to the following factors: 'efficiency in patient care', 'cost reduction', 'capabilities to improve practice' and 'security and confidentiality'. However, Estai's proposed four factors for the perception of TDS were as follows: 'the dentist's perceived usefulness of Teledentistry', 'the efficiency of Teledentistry', 'the benefits of Teledentistry for patients' and 'the practice-related use of information technologies'.

In the context of teledentistry, the theoretical technology acceptance model (TAM), proposed by *Davis (1989)*, plays a crucial role in understanding the factors influencing the acceptance and adoption of technology by dental practitioners (*Kamal, Shafiq & Kakria, 2020*). The results of our EFA revealed that the TDS-20 questionnaire, a modified version of the TDS-24, effectively captured four essential factors related to teledentistry acceptance. These factors, namely 'efficiency in patient care,' 'cost reduction,' 'capabilities to improve practice,' and 'security and confidentiality,' align with the TAM constructs, shedding light on how dental practitioners in Saudi Arabia perceive the ease of using teledentistry and its usefulness in their practice. These results align with findings from multiple studies (*Aboalshamat, 2020*; *Alsharif & Al-harbi, 2020*; *Almazrooa et al., 2021*). Furthermore, the study's findings indicate the importance of considering cultural and contextual differences when applying TAM to teledentistry. The variations in factor loadings and factor naming compared to Estai's proposed factors highlight the need for a nuanced approach in understanding technology acceptance within specific regions and populations. This reinforces the significance of conducting cross-cultural validation studies and tailoring technology acceptance models to the unique needs and perspectives of diverse user groups.

The strengths of this study are that it is likely the first study to cross-culturally validate the TDS using a diverse sample in a large-scale cross-sectional survey. This study introduces a refined survey version named TDS-20, which is a modification of the original TDS-24 questionnaire. This adaptation stems from the results of an exploratory factor analysis, which unveiled that items 12, 16, 21, and 24 exhibited low factor loadings and were subsequently excluded from the TDS-24. These exclusions were made in TDS-20 to enhance the scale's precision and relevance in the context of our research. This separate study goes beyond the surface by delving into the intricacies of the survey, employing exploratory factor analysis. The results unveil intriguing findings as the proposed domains in the survey underwent significant shifts when applied in the Saudi context, suggesting the emergence of new domain categories.

The limitations of the study can be summarised as follows: the information was retrieved from the participating subjects using the English language, which is the language of teaching in dental schools in Saudi Arabia. However, cross-cultural revalidation in other languages is warranted. A further limitation of this study is its exclusion of clinicians who are not active on social media networks. One should consider that the factor structure of the TDS could have been influenced by the present sample data (*i.e.*, those who were interested in the study participated). Therefore, revalidation studies using different populations and random samples are needed to overcome sampling bias and confirm the latent variable structure. Participants' self-reported perceptions were not verified with

objective measures and, as such, were subject to recall bias. Finally, this study's sample had socially advantageous backgrounds. Saudi Arabia is classified as a high-income country, but the revalidation of the TDS in low- and middle-income countries should be undertaken. Our study was also limited by the cross-sectional design and use of an anonymous online questionnaire, which made it challenging to assess test-retest reliability.

In conclusion, the TDS was found to be valid and reliable in assessing teledentistry perceptions among dental practitioners in this study sample. This study proposes a modified version of the TDS-24 survey as the TDS-20. Considering the findings from this study, it is recommended that the modified TDS-24 survey can guide further development of the survey to better suit resemble context. The results of the factor analysis provided support for the underlying structure of the proposed TDS. While most items demonstrated satisfactory loading values, the exclusion of three items and the cross-loading of others highlighted potential areas for improvement. The factors of efficiency and cost exhibited good internal consistency, whereas the capability and security factors showed room for improvement. The findings derived from this factor analysis contribute to our understanding of the proposed TDS and its potential applications. However, this scale can be improved if the items are adjusted *via* further studies. Further construct validation and construct convergence validation of the TDS with unexplored factors relevant to teledentistry use (*e.g.*, technology-related items) should be explored in future research to enhance scale validation.

### Funding
The authors received no funding for this work.

### Competing Interests
The authors declare that they have no competing interests.

### Author Contributions
- Alla Alsharif conceived and designed the experiments, performed the experiments, prepared figures and/or tables, authored or reviewed drafts of the article, and approved the final draft.
- Doaa Felemban analyzed the data, authored or reviewed drafts of the article, and approved the final draft.
- Hala Bakeer analyzed the data, authored or reviewed drafts of the article, and approved the final draft.
- Saba Kassim conceived and designed the experiments, performed the experiments, analyzed the data, prepared figures and/or tables, authored or reviewed drafts of the article, and approved the final draft.

## Human Ethics

The following information was supplied relating to ethical approvals (*i.e.*, approving body and any reference numbers):

Ethical clearance to conduct this study was obtained from the Research Ethical Committee at Taibah University College of Dentistry.

## Data Availability

The data is available in the Supplemental File.

## Supplemental Information

Supplemental information for this article can be found online at http://dx.doi.org/10.7717/peerj.16834#supplemental-information.

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
