# Peer review of "Validity and reliability of a teledentistry survey among dental practitioners in Saudi Arabia"

_PeerJ, doi:10.7717/peerj.16834_

## Round 0.1 · original submission · Major Revisions

In light of reviewers' criticisms, the article must be subjected to major revisions. The discrepancy between the survey's name (TDS-24) and the final reported version (TDS-20) should be clarified .

Reviewer 1 ·

Basic reporting

Teledentistry has been covered extensively in the region

I do not see any additional information provided in this manuscript

Experimental design

I have reservations on how this study was conducted

A 10 minutes survey is impossible to complete (time wise)

The number of participants is beyond sample size calculation

Validity of the findings

I have reservations on results of this study

·

Basic reporting

In this study conducted to establish the psychometric properties of TDS-24 scale by Estai et al., among dentists in Saudi Arabia, the authors have presented a commendable manuscript in terms of methods, results, their interpretation and the novelty of the research question. However, few important points need to clarified or modified in the manuscript:
Introduction: How are the dentists at Saudi Arabia different than the dentist populations in other countries, where the TDS-24 has been studied? This point needs to be elaborated to justify the need for the study.

Experimental design

The methods section is well written and clearly describes all aspects of data collection.

Validity of the findings

Results:
1. How were the four factors in TDS-24 arrived at based on factor analysis?
2. The reasons for excluding items 12, 16, ,21 and 24 are not clear. Factor loadings for items 16,21 and 24 are missing. No data is provided for item 24. Can the authors demonstrate the change in the psychometric analyses results with and without these items to justify their omission?

Additional comments

Conclusion: Are authors suggesting a 20 item, modified TDS-24? The same needs to be clearly mentioned. Potential modification for the scale during future use should be specifically suggested.

Reviewer 3 ·

Basic reporting

This review on a study conducted in Saudi Arabia to assess the validity and reliability of the TDS-24, a teledentistry survey, among dental practitioners.
The abstract clearly states the objective of the study, which is to examine the validity and reliability of the TDS-24 survey.
The methods used in the study, including the questionnaire, sample size, and recruitment method, are well-explained. It mentions the use of exploratory factor analysis (EFA) and Cronbach's alpha for assessment.
The results abstract presents the key findings of the study, including the identification of a four-factor structure and the corresponding Cronbach's alpha values for each factor. This information provides insight into the psychometric properties of the survey.
The conclusion is well written and by confirms that the TDS-20 (not TDS-24 as initially mentioned) demonstrated adequate psychometric properties, making it a valid and reliable tool for assessing perceptions of teledentistry among dental practitioners in the study.
Discussion provided a brief explanation of the four identified factors for better understanding.

Experimental design

no comment

Validity of the findings

no comment

Additional comments

Overall, the article effectively communicates the study's objectives, methods, and key findings. However, there seems to be a discrepancy between the survey's name (TDS-24) and the final reported version (TDS-20), which could be clarified for readers.

Reviewer 4 ·

Basic reporting

The paper is well-presented, the study rationale is clear and includes a relevant literature review.

Consider an existing conceptual framework such as TAM or diffusion of technology.

Experimental design

The study design used is suitable and appropriate to address the study objectives:
Further info is required as per the following:

• Noted that the survey adapted from Estai et al and Mandal was used in the current study without further development. The mandal survey was developed at about 18 years old, which is outdated and missing many items related to users’ behaviours towards technology such as ease of use and predicted usage (refers to Technology Acceptance Model). It might be too late to consider further questions in the survey but it would be interesting to discuss or address this.
• The recruitment process is concise, please elaborate more on this. More info about inclusion and exclusion criteria.
• It is unclear if the survey was distributed in Arabic language.
• What are the outcome measures that you tried to assess?
• More info is warranted about data management

Validity of the findings

Findings are valid and accurate: consider further data analysis and more findings such as:

• Consider assessing test-retest reliability: This involves administering the same survey to the same group of respondents at two different points in time and then calculating the correlation between the two sets of responses.
• Consider a correlation matrix to evaluate the construct validity of your survey by examining the relationships between the different theoretical constructs or variables you're measuring.

Additional comments

Comments to authors:
The present study aimed to examine the validity and reliability of the TDS-24 in a sample of dental practitioners in Saudi Arabia.

• Noted that the survey adapted from Estai et al and Mandal was used in the current study without further development. The mandal survey was developed at about 18 years old, which is outdated and missing many items related to users’ behaviours towards technology such as ease of use and predicted usage (refers to Technology Acceptance Model). It might be too late to consider further questions in the survey, but it would be interesting to discuss or address this.
• Consider an existing conceptual framework such as TAM or diffusion of technology.
• The recruitment process is concise, please elaborate more on this. More info about inclusion and exclusion criteria.
• It is unclear if the survey was distributed in Arabic language.
• What are the outcome measures that you tried to assess?
• More info is warranted about data management
• Consider assessing test-retest reliability: This involves administering the same survey to the same group of respondents at two different points in time and then calculating the correlation between the two sets of responses.
• Consider a correlation matrix to evaluate the construct validity of your survey by examining the relationships between the different theoretical constructs or variables you're measuring.
• Discussion: what is the implication of findings and future recommendations

---

## Round 0.2 · Minor Revisions

The quality of the manuscript has improved significantly compared to the last submission. The manuscript must be revised according to the reviewers' criticisms before being published.

·

Basic reporting

I appreciate the authors for their response to the reviewer comments in a specific and relevant manner.

Experimental design

The authors have improved the clarity in the methods section by providing details on inclusion and exclusion criteria and the sample size.
Line 145-151 elaborates on the objective and hence can be shifted to the objective section of the article (at the end of introduction)

Validity of the findings

The results are now better presented explaining the need for change from TDS-24 to TDS-20

Additional comments

nil

Reviewer 4 ·

Basic reporting

See comments below

Experimental design

See comments below

Validity of the findings

See comments below

Additional comments

Thank you for your revision, the manuscript has been improved; however, there are some comments that haven't been adequately addressed:

Consider discussing findings in light of theoretical frameworks such as TAM or diffusion of technology. They may help to explain users' behaviors towards the use of teledentistry. The authors already mentioned that this has been addressed in the manuscript, but no information exists.

Consider highlighting the recruitment process of participants or dentists, for example, how potential participants were invited or approached. Was this through social media or email registry? The authors just added info about the inclusion criteria.

Please highlight the main results for the correlation matrix in a table.

---

## Round 0.3 · Major Revisions

Some situations must be addressed before the article can be considered for publication.

English language needs to be carefully revised.

In page 3 line 89 United Kingdom is mentioned twice.

The penultimate sentence of the introduction should be revised.

In line 190 add “DECLARATION “of Helsinki.

Regarding the sampling process, a doubt arises. If the link to the questionnaire was disseminated through social media (Twitter, Facebook, etc.) how can the authors guarantee (as they claim) that "800 TDS questionnaires were distributed"

In the limitations, it should be mentioned the fact that the study excludes clinicians who do not use social networks.

The authors need to clarify better how the methodology used allows reaching this conclusion: "The TDS was found to be valid and reliable in assessing teledentistry perceptions among dental practitioners in this study sample".

---

## Round 0.4 · accepted · Accept

After revisions, the manuscript is ready to be considered for publication.